# A Triple-Band Hybridization Coherent Perfect Absorber Based on Graphene Metamaterial

**Xinpeng Jiang [1,2], Zhaojian Zhang [1,2], Kui Wen [1,2], Guofeng Li [1,2], Jie He [1,2] and Junbo Yang [2,*]**

1    College of Liberal Arts and Sciences, National University of Defense Technology, Changsha 410073, China;
     jackson97666@163.com (X.J.); 376824388@alumni.sjtu.edu.cn (Z.Z.); kuiwen93@hotmail.com (K.W.);
     liguofeng24@163.com (G.L.); 18795898068@163.com (J.H.)
2    Center of Material Science, National University of Defense Technology, Changsha 410073, China
*    Correspondence: yangjunbo@nudt.edu.cn



**Featured Application: We applied "Lumerical FDTD Solution" in the simulation process, which develops high-performance photonic simulation software, enabling designers to predict light's behaviour within complex structures and systems. And this software is constantly being update to improve functionality, especially for the simulation of new materials such as graphene, $Si_3N_4$ and so on.**

**Abstract:** In this paper, a triple-band hybridization coherent perfect absorber based on graphene metamaterial is proposed, which consists of graphene concentric nanorings with different sizes and a metallic mirror separated by $SiO_2$ layer. Based on the finite-difference time-domain (FDTD) solution, triple-band coherent perfect absorption is achieved at frequencies from 0.6 THz to 1.8 THz, which results from the surface plasmon resonance hybridization. The wavelength of the absorption peak can be rapidly changed by varying the Fermi level of graphene. Most importantly, the wavelength of the absorption peak can be independently tuned by varying the Fermi level of the single graphene nanoring. Moreover, the triple hybridization perfect absorber is angle-insensitive because of the perfect symmetry structure of the graphene nanorings. Therefore, our results may widely inspire optoelectronic and micro-nano applications, such as cloaking, tunable sensor, etc.

**Keywords:** terahertz; graphene; hybridization; perfect absorber

## 1. Introduction

Metamaterials have unique advantages for the regulation of electromagnetic waves. It is a powerful structure formed by the periodic arrangement of many micro-nano structures, such as concentric rings [1,2], nano-rods [3,4] and the metal split-ring resonators [5,6]. Metamaterial absorbers have received extensive attention from researchers since Landy et al. [7] proposed metamaterial-based perfect absorber (MPA) in 2008. The work based on metamaterials has promoted the development of many functional devices, such as optic cloaks [8,9], sensors [10], optical switches [11] and so on. However, most of these proposed MPAs still have a strict working bandwidth. It is hard to change the structure in integrated devices.

Graphene, as a two-dimensional material with many excellent properties, is widely used in the design of metamaterials. The graphene nanoring or nanodisk is used to realize the functions of light modulation, switching and sensing in 2013 [12]. Graphene multilayer wave absorbers [13–15] have been proposed many times, but the fabrication of multilayer structures is complex and multilayers cannot be guaranteed. Therefore, much research has conducted extensive explorations on single-layer graphene structures [16–18], whereas most graphene metamaterials have narrow bands or have very limited

frequency adjustments, especially the frequency of adjustment is not flexible enough. The frequency changes at the same time, which greatly limits the application of graphene metamaterials. Researchers have done many experiments to explore graphene metamaterial absorbers, Fang Z. et al. utilize an array of closely packed graphene nanodisks, realizing absorption efficiency from 3% to 30% in the infrared region of the spectrum [19]. In addition, a cavity-coupled nanopatterned graphene absorber has achieved high absorption theoretically and experimentally between 8–12 μm, and the adjustable range is about 2 μm [20]. These experiments show that the graphene absorber is technically feasible. Therefore, we have proposed a graphene absorber that can effectively control a single frequency.

In this work, we propose a triple-band graphene structure that can be adjusted separately. Our device is a sandwich structure which consists of a single-layer of graphene concentric triple ring (GCTR) array and a metallic mirror separated by SiO₂ layer. Compared with the previous work of others before [21], our device has a wider range of adjustment freedom, and our proposed absorber can achieve a Tri-band and Dual-band conversion. FDTD is used to simulate the response frequency of 0.5–2.5 THz, and the single-frequency or multi-frequency conversion can be achieved by separately adjusting the Fermi level of a single pattern. In addition, we use a hybridization model to explain the coupling of the graphene rings in a proposed absorber.

## 2. Structure and Method

The schematic of the structure is shown in Figure 1. The proposed structure is composed of a single layer of patterned graphene, a layer of silicon dioxide as a loss layer and a mirror layer of perfect electrical conductor (PEC). Among them, the thickness of $SiO_2$ is $t = 28$ μm, which is considered the permittivity $\varepsilon = 3.9$, and the thickness of PEC is $h = 3.5$ μm. In order to control the Fermi level of the top graphene, we set the Au electrodes as the top gate and a patterned ion-gel above the graphene, and a conductor below the graphene. We patterned the ion-gel into a three-layered split-ring structure in Figure 1. Each layer of the split-ring was connected to an Au gate. Thus, three Au gates were used to separately adjust each layer of graphene. The size of the ion-gel was similar to the graphene's three-ring pattern. We hope to use this method which the Fermi level is dynamically adjustable to achieve many kinds of absorption patterns in one absorber. The ion-gel pattern which was omitted to simplify the calculation is designed to discretely adjust the three graphene nanorings [22,23]. We also provide some evidence in Figure S1 to prove this point. And some researchers believe that the phonons between the ion-gel and oxide silica will affect the absorption of the absorber [12,24].

In addition, as shown in Figure 1, we give the results of the graphene patterning process. We have designed a pattern of three concentric rings of graphene. On the one hand, this pattern is a good perfect symmetrical structure, which can solve the problem of polarization sensitivity. On the other hand, through the regulation of the three rings and the mutual coupling between the three rings, we can achieve a variety of functions, such as continuous perfect absorption and mode conversion of the absorption frequency. In the process of FDTD simulation, we set the convergence time $\tau = 1$ ps, $\mu = 2 \times 10^4$ cm²·V⁻¹·s⁻¹ (there is a corresponding relationship between $\tau$ and $\mu$). The design of graphene's electron-phonon relaxation time can be achieved by using the mechanical exhalation method in high-quality graphene in other work [25,26].

In our work, we give $E_f = 0.7$ eV of any initial value, and obtain the simulated data by scanning the values of different Fermi levels without changing the structural parameters. We set override x mesh and y mesh is 5 nm and override z mesh is 1 nm. The size of mesh is much smaller than the minimum size of the pattern to ensure the accuracy of the absorption and the using of 1 nm in the setting of the z-axis mesh is to ensure that the single-layer graphene structure is well scanned. In addition, according to the data results from FDTD—the reflectivity $R(\omega)$ in the upper space and the transmittance $T(\omega)$ in the bottom space, we can get the result of the absorption rate of the absorber $A(\omega) = 1 - R(\omega) - T(\omega)$, this method is widely used by researchers [13,17–19].

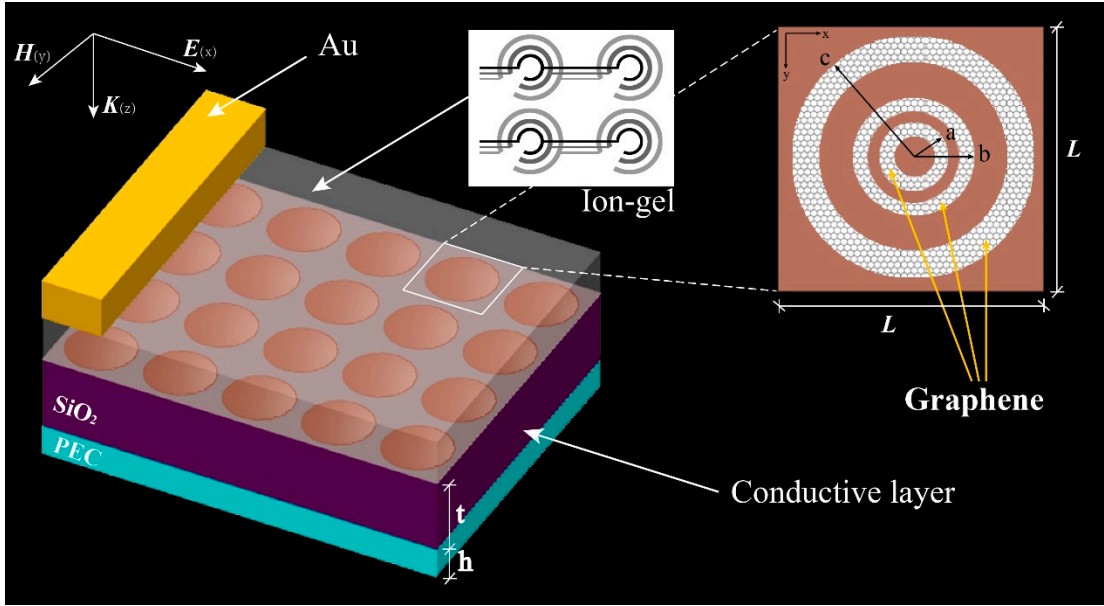

**Figure 1.** Schematic of the proposed broadband metamaterial-based perfect absorber (MPA)consisting of the graphene concentric triple ring (GCTR) and the perfect electrical conductor (PEC) mirror separated by a thin $SiO_2$ spacer. The ion-gel was spin-coated on the graphene nanostructures and contacted to the Au electrodes as the top gate. Also, an ultra-thin and transparent conduct thin layer was deposited between the $SiO_2$ and graphene as the bottom gate. The geometrical parameters of the proposed MPA are $t = 28$ μm and $L = 11$ μm. The outer radii of three rings are $a = 1.5$ μm, $b = 2.5$ μm, $c = 4.5$ μm, the widths of three rings are 0.2 μm, 0.5 μm, 1.1 μm.

Regarding the conductivity tunability of graphene, it can be known from the Kubo formula [27] that the conductivity is determined by the internal and external conductivity [28–30],

$$\sigma(\omega) = \sigma_{intra}(\omega) + \sigma_{inter}(\omega) \tag{1}$$

$$\sigma_{intra}(\omega) = \frac{2e^2 k_B T}{\pi \hbar^2} \frac{i}{\omega + i/\tau} \ln\left[2\cosh\left(\frac{E_f}{2k_B T}\right)\right] \tag{2}$$

$$\sigma_{inter}(\omega) = \frac{e^2}{4\hbar^2}\left[\frac{1}{2} + \frac{1}{\pi}\arctan\left(\frac{\hbar\omega - 2E_f}{2k_B T}\right) - \frac{i}{2\pi}\ln\frac{\left(\hbar\omega + 2E_f\right)^2}{\left(\hbar\omega - 2E_f\right)^2 + 4(k_B T)^2}\right] \tag{3}$$

where $\omega$ is the frequency of the electromagnetic wave, $E_f$ is the chemical potential (Fermi level), $\tau$ is the electron-phonon relaxation time, and $T$ is the ambient temperature. Also, $k_B$ is the Boltzman constant, $e$ is the electron charge and $h$ is the Plank constant, respectively. The surface conductivity can be safely reduced to a class of Drude models [31–33].

$$\sigma(\omega) = \frac{e^2 E_f}{\pi \hbar^2} \frac{i}{(\omega + i/\tau)} \tag{4}$$

The principle of graphene conductivity tunability provides the possibility to modulate the work bandwidth of the device, and its Fermi energy can be freely adjusted over a wide range by applying an electrostatic bias [33].

We apply the quasi-static description [34,35] of graphene tunable conductance theory to analyze the surface plasmons generated by bound electrons. From previous work [35–37], we know that this method can get a good approximation. The operating frequency of the system generating surface

plasmons is determined only by the geometry and dielectric function. Under the condition of phase matching, we can get

$$\omega_p = \frac{e}{\hbar}\sqrt{\frac{E_f}{\pi\eta\varepsilon W_{eff}}} \tag{5}$$

where $\omega_p$ is the angle frequency of surface plasmon resonant frequency, $\varepsilon$ is the average dielectric constant which can be simply considered as the average $\varepsilon$ of the upper layer $\varepsilon_1$ and the lower layer $\varepsilon_2$, $W_{eff}$ is the effective ribbon width and $\eta$ is the dimensionless parameter, which uniquely determines the electrodynamic responses of the nanoribbon array. The Fermi level of graphene has a linear relationship with the carrier concentration of graphene

$$E_f = \hbar v_F n_s \tag{6}$$

where Fermi velocity and Planck constant are constant values, $V_f = 10^6$ m s$^{-1}$. We can control the carrier concentration in graphene by changing the gate voltage. From the simple capacitor model and the linear relationship between the carrier concentration of graphene and the external gate voltage, the equation can be expressed as [38,39]

$$n_s = \frac{\varepsilon\varepsilon_0|V_g - V_{Dirac}|}{ed}. \tag{7}$$

Among them, $\varepsilon$ is consistent with the previous expression as the relative dielectric constant of SiO$_2$, $\varepsilon_0$ and $d$ represent the vacuum dielectric constant and the thickness of the insulator layer. $|V_g - V_{Dirac}|$ is the amount of external gate voltage applied. In this way, we can fully understand the process of the proposed absorption structure changing the Fermi level, and this process will qualitatively change the surface plasmon frequency, which will affect the operating frequency of the absorber. Since single-layer graphene is an extremely thin material, we can think of it as an ultra-thin film, which has a certain equivalent dielectric constant:

$$\varepsilon_g = 1 + \frac{i\sigma}{\omega\varepsilon_0 b}. \tag{8}$$

Here, $\varepsilon_0$ and $\omega$ respectively represent the dielectric constant of the vacuum and the frequency of the incident wave, which have been given in the previous equation, $b$ represents the thickness of graphene. According to $n = \sqrt{\varepsilon_g\mu_g}$, the refractive index of graphene can be obtained. Where $\varepsilon_g$ is equivalent dielectric constant and $\mu_g$ is independently specified [40,41]. In this equation, we can know that conductivity affects the dielectric constant of graphene films. In addition, the electrical conductivity will directly affect the characteristics of the surface plasmon polarization (SPP) excited by graphene [42,43].

## 3. Results and Discussion

### 3.1. The result of the Structure Tuning

The working principle of this reflective MPA can be understood as follows. When the electromagnetic wave illuminates the device, the SPP is excited on the top graphene layer and the redundant energy in the absorber was reflected by the bottom PEC mirror, resulting in strong magnetic resonance [44–47]. It is known from the properties of SPP that the SPP excited by the MPA will change the direction of energy propagation, thereby achieving the loss of these energy at the surface without reflection. Earlier works [22,25,48,49] showed that a local surface plasma mode or a gap plasma mode illuminated by incident light will be excited inside the graphene microstructure, resulting in a significant enhancement of light absorption.

First, with the above theory and the Formula (3), we can know that by changing the Fermi level of graphene, we can change the response frequency of the graphene absorption structure, which will

change the position of the absorption peak of the graphene absorption structure. We control the Fermi level by changing the voltage between the gates. The relationship of the gate-voltage and the Fermi level is shown in the Formulas (6) and (7). Utilizing this property, we tune the Fermi level of the GCTR. The results are shown in Figure 2.

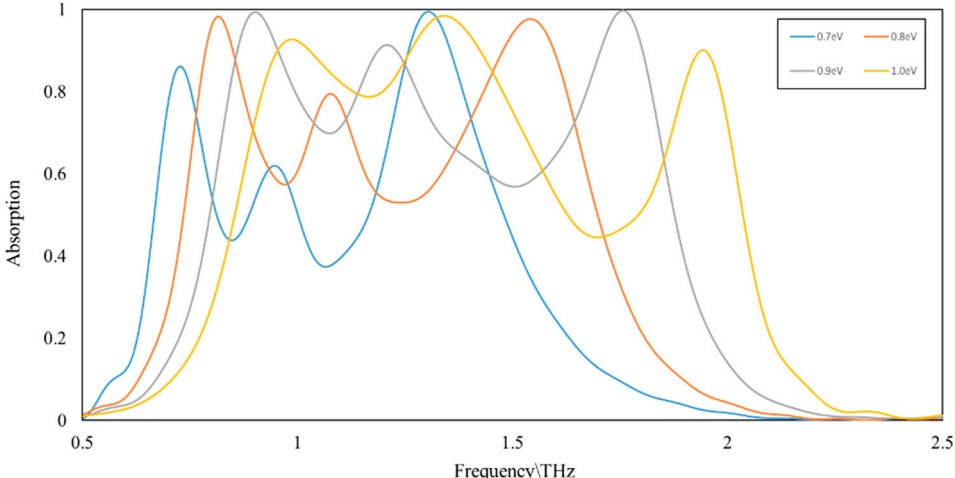

**Figure 2.** The simulated absorption spectrum of the absorber which the Graphene at the different Fermi levels (blue line is $E_f = 0.7$ eV, orange line is $E_f = 0.8$ eV, gray line is $E_f = 0.9$ eV, yellow line is $E_f = 1.0$ eV).

The Figure 2 depicts the simulated absorption spectrum of the absorber which the Graphene at the different Fermi levels. From the theory of adjustable conductivity, we finally give the approximate relationship between the SPP response frequency $\omega$ (the frequency corresponding to the absorption peak) and the Fermi level $E_f$ of graphene. It can be seen that under the control variables, without changing any structural parameters of the absorption structure, as the graphene Fermi level increases, the surface plasma response frequency increases. This is consistent with the absorption spectrum we obtained through FDTD simulation. When we adjusted the graphene three-ring Fermi level $E_f$ from 0.7 eV to 1.0 eV, the peak of the absorption blue shift (the surface plasmon response frequency increase). As the Fermi level increases, we can draw a general conclusion that all the graphene absorbing structures appear as a blue shift. It is consistent with the conclusion of conductance regulation. Moreover, the absorptions of the absorption peaks are fluctuating between 61% to 98% (the second peak) with the changes of the Fermi level. As well as that, it is apparent that the Fermi level plays a significant role in the absorption and the frequency of the absorption peak.

Cause to the indispensable role of the Fermi level, we have a new idea that we maintain the Fermi level of the two rings unchanged and change the Fermi level of the single graphene ring (SGR). In this way, we can achieve the adjustment of SGR and thus achieve higher continuous absorption efficiency of the absorption peak with a poor coupling effect. The results are shown in Figure 3.

The Figure 3 depicts that SGR adjustment not only can change the response frequency of the Graphene but also can change the efficiency of the absorption peak which will tune the absorber without the structure reengineering. Afterwards, we can see that we increase a new perfect absorption by adjust the Fermi level of SGR. Using the theory of adjustable conductivity, we finally provide the approximate relationship between the surface plasmon response frequency $\omega$ (the frequency corresponding to the absorption peak) and the Fermi level $E_f$ of graphene. Using this relationship, we can adjust the Fermi level of the largest graphene ring from 1.0 eV to 0.7 eV, and we can find that the absorption peak has a red shift phenomenon, which is consistent with the tunable conductivity model. We also found that the regulation of a single Fermi level can achieve the intersection of two absorption peaks to produce the sum peak effect. The absorption of the third absorption peak increased from 75% to 99% due to the coupling between the innermost ring and the middle ring. Furthermore, using this effect, we

expected to achieve the conversion from three frequencies to dual frequencies. In a way, we achieved the switch between triple-band to dual-band by fluctuating the SGR. We can therefore develop a true multi-function by using the Fermi level adjustment of the different sizes of the graphene ring.

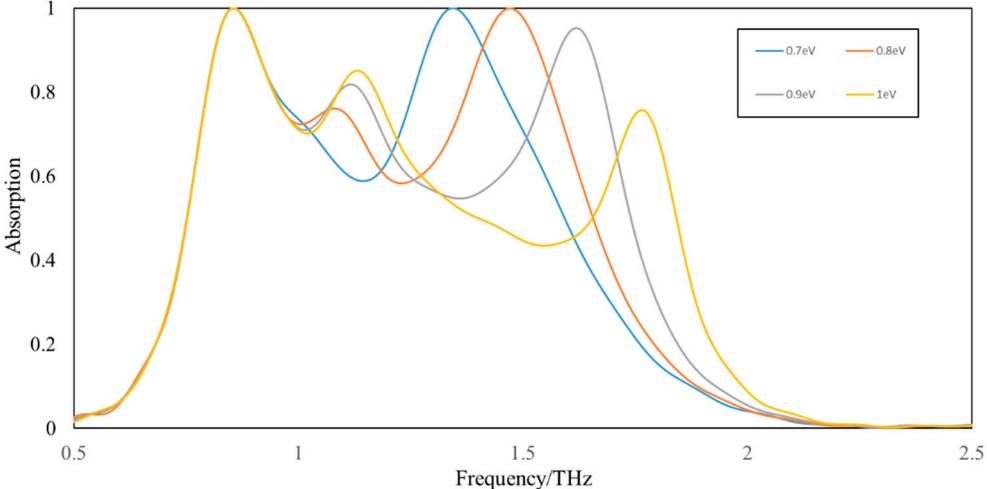

**Figure 3.** The simulated absorption spectrum of the absorber which the two rings' Fermi level is constant (1.0 eV) and the smallest SGR's Fermi level is varying from 0.7 eV to 1.0 eV. (blue line is 0.7 eV, red line is 0.8 eV, gray line is 0.9 eV, orange line is 1 eV).

Of course, in the tunable conductance model, we focus more on the influence of the regulation of the Fermi level on the response frequency. Before the Fermi level adjustment, we hope to design an absorber with a good absorption in the work frequency. Therefore, we started a discussion on size adjustment and found out the relationship of the size and absorption spectrum.

We keep the Fermi level and other structural parameters unchanged, and the size of the outer ring is from 0.9 μm to 1.5 μm, with 0.2 μm as an interval. We also obtained a lot of interesting results, which are shown in Figure 4.

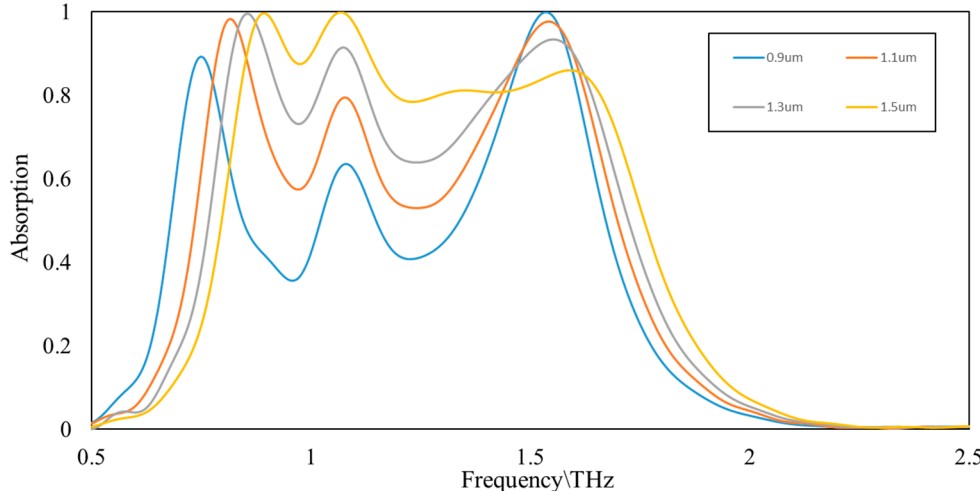

**Figure 4.** The simulated absorption spectrum of the absorber which the two rings' sizes are constant (0.3 μm and 0.5 μm) and the biggest SGR's size is varying from 0.9 μm to 1.5 μm. (The blue line is 0.9 μm, red line is 1.1μm, gray line is 1.3 μm and orange line is 1.5 μm).

We can see that there is a transformation from 0.9 μm to 1.5 μm, which is the frequency of the perfect absorption changes (the variation of the absorption peak). We achieve a high coherent

absorption which can maintain an absorption rate above 70% from 0.8 THz to 1.7 THz. Also, the proposed MPAs can realize perfect absorption at the 0.89 THz and at the 1.07 THz rates. The frequency of this two absorption peak is hardly affected. Moreover, we also can provide a conclusion that a single ring can have an entirely different effect on different rings in the proposed absorber. For the middle ring, the coupling can make the absorption enhance. For the inner ring, the coupling make the absorption reduce. The effect of the absorption can be explained by the hybridization model.

### 3.2. Theory Testify

Next, we performed a full wave simulation to verify the hybridization model used in the proposed graphene absorber structure. Based on the size changing in Figure 4, we provided an analysis of the absorption variations of the absorption peak of the three-ring system, for which sizes of the graphene rings are 0.2 μm, 0.5 μm and 1.1 μm based on the hybridization model. For the full wave simulation, we selected images with frequencies at three absorption peak positions to observe the superposition of the absorption structure. By changing the size of the out SGR, we can clearly find that the coupling hybrid process of the inner double ring and the out ring respectively produces two types of coupling: strong coupling and weak coupling, which directly leads to changes in the absorption peak. In addition, we do not change the outer size of the outer ring (without changing the size of the unite cell), and change the coupling by changing the inner size of the out SGR. When the inner size is reduced (the outer ring size becomes larger), the coupling effect is enhanced, which in turn makes the strong coupling effect stronger and the weak coupling effect weaker, meaning the changes of the two absorption peaks are opposite. Our results are perfectly explained in the hybridization model theory of others [12,21,50,51] discussed previously.

Figure 5 depicts the full wave simulation and the electric field distribution at three absorption peak which can perfectly describe the coupling of the structure. The full-wave simulation method is used to simulate the contrast of the field strength. The color for which the field is strong is reddish and the contrast is intensive, while the color being blue means the field is weak. Based on the pointing of the magnetic needle and the full-wave simulation which the FDTD gives, we give the magnetic needle schematic diagram, which can represent the direction and strength of the field. The direction of the magnetic needle represents the direction of the electric field from strong to weak, and the size of the magnetic needle indicates the strength of the field strength at different positions. According to the direction of the magnetic needle, we can give the approximate distribution of the strength of the electric field and the corresponding positive and negative electrodes. We mark "+" "−" on the edge of the ring using by the magnetic needles. The actual SPP electric field in the ring should remain neutral. Therefore, the up mark and the low mark are symmetrical, so that the total electric field of the ring is neutral. Also, both sign of edges of a ring is identical, so we use a sign ("+/−") to represents the electric field distribution near the ring.

Based on the full wave simulation, there is no doubt that the three graphene rings coupling at all of the absorption peaks. Utilizing the control variable method, we simulated the full-wave simulation states at different frequencies (two absorption peak positions which Figure 5a is 1.07 THz and Figure 5b is 1.54 THz.) under the same model (the original structure is given in Figure 1), and found that there are two modes of full-wave simulation at two different frequencies. Furthermore, it is explained that changing the size of a single graphene ring structure (the result shown in Figure 4) will cause the two absorption peaks to have different coupling directions due to the existence of coupling, and the change direction of the absorption effect is the same as before. [12,21,50,51] In the electric field distribution, we can find that the electric field is "+ + − + − −" in the MPA at the 1.07 THz. Also, the electric field is "+ − + − + −" at the 1.54 THz. This perfectly proves the hybridization model theory. Because of the different hybridization modes, the absorption peaks present the different variation trends. As the coupling effect strengthened, the peak of 1.07 THz increased significantly (strong coupling), while the peak of 1.54 THz weakened (weak coupling). We have perfectly deduced these two hybridization modes through our proposed triple-band perfect absorber, and these two hybridization modes can

provide us with design inspiration at the beginning of structural design. This inspiration made us get the proposed coherent perfect absorber at the beginning of the design, so that we can adjust the working frequency of the absorber separately or uniformly. In addition, our research may extend the hybridization model theory.

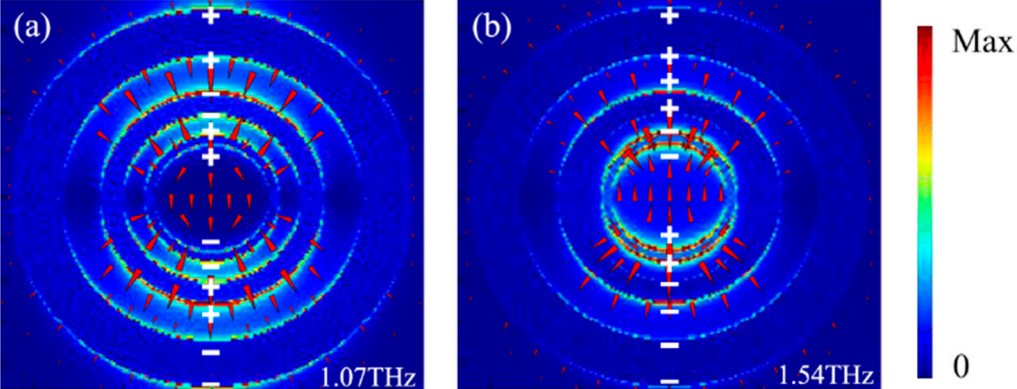

**Figure 5.** Simulated electric field patterns (arrow for the direction and color for the contrast of the field strength) at the top graphene layers of the MPAs in Figure 3. Red line at the absorption peak. (**a**) is the peak at the 1.07 THz and (**b**) is the peak at the 1.54 THz.

Finally, it is clear that our structure is an angle-insensitive structure. Due to the perfect symmetry of the graphene nano-ring, when we turn the direction of the TE wave, but do not change the direction of the electromagnetic wave, it is actually only the corresponding rotation of the coupling position that changes, without affecting absorption.

### 3.3. The Discussion of Polarization-Angle-Insensitive and Angle-Insensitive

Based on the perfect symmetrical pattern of graphene nanorings, we discuss the polarization-angle influence of the proposed triple hybridization absorber. We set up structural parameters like the initial value in Figure 1 and fix $E_f$ = 0.9 eV to all of the three rings. Then we vary the polarization from 0 degrees to 45 degrees at 15-degree intervals. Figure 6a shows a schematic diagram of the angle of incidence relative to the absorption structure, specifically indicating the position of the polarization angle $\varphi$. We can clearly find that the data graph of the absorption effect is strictly insensitive to the change of the angle in Figure 6b. The absorption graphs of 0 degrees, 15 degrees, 30 degrees and 45 degrees are completely coincident. The slight fluctuations of the results obtained in FDTD can be ignored. Therefore, we can consider that our perfect absorption structure is an absorption structure that is not sensitive to the polarization angle, which is crucial for the application of the absorption structure in practical devices.

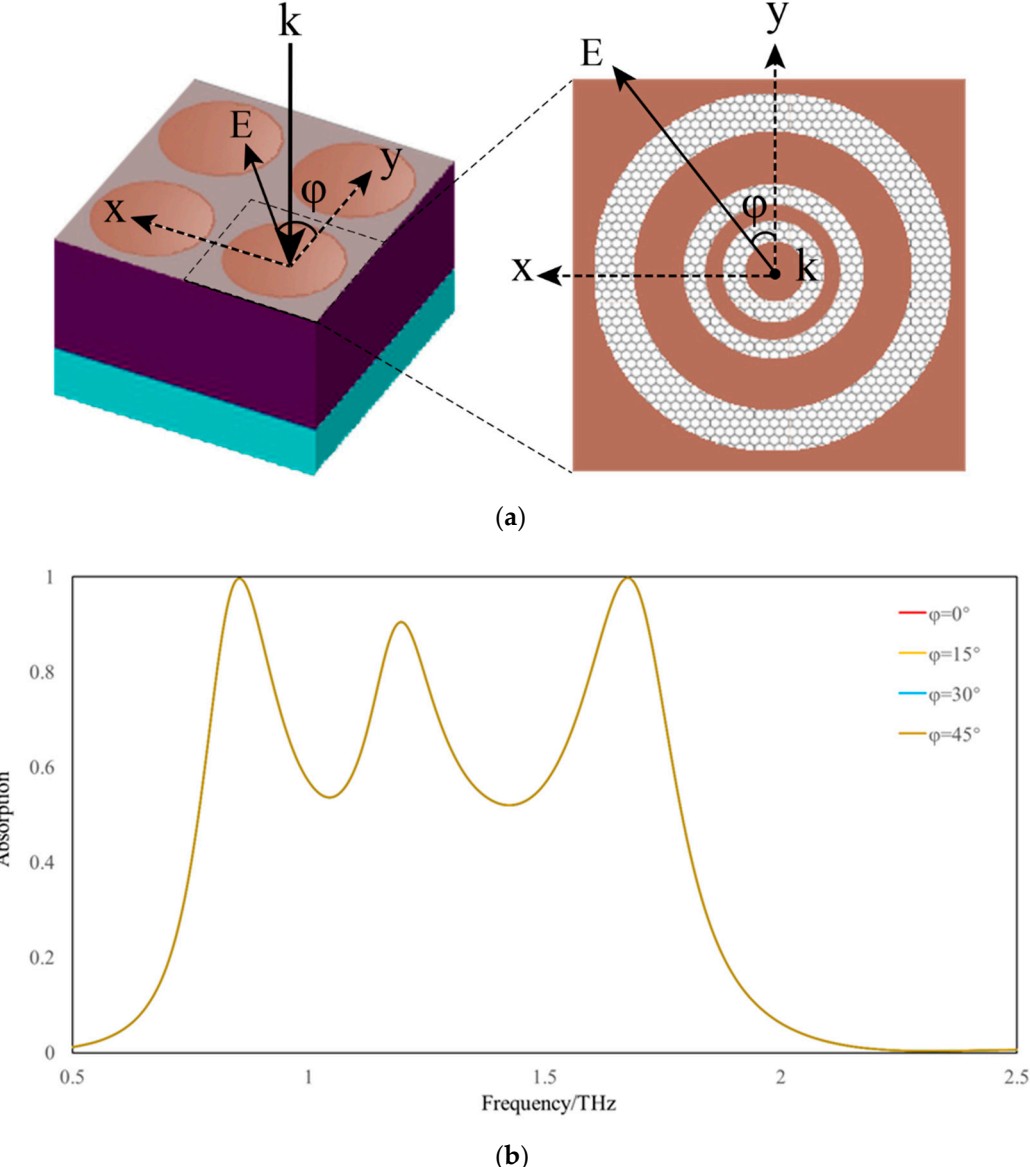

(a)

(b)

**Figure 6.** Schematic diagram and absorption of the polarization angle of the incident electromagnetic wave propagating to the absorbing structure. (**a**) is the scheme which describes the polarization angle $\varphi$ to the proposed absorber, (**b**) is the absorption of the different polarization angle in the range of 0.5 THz to 2.5 THz. The red line indicates the absorption rate at $\varphi = 0°$, the yellow line indicates the absorption rate at $\varphi = 15°$, the blue line indicates $\varphi = 30°$, and the brown line indicates $\varphi = 45°$.

We also tried to simulate the incident angle $k$ from 0° to 30° every 10° to get a graph of the absorption data for the incident angle change (We define the angle of $k$ as the angle between the propagation direction of the incident light and the normal to the plane of the absorption structure). In FDTD, we set the $E_f = 0.8$ eV and the structure keeps original parameters which we give in Figure 1. From Figure 7, we can see that the response frequency has a certain change with the angle. Fluctuations occur, but generally does not affect the absorption effect of the entire absorption structure. Since the proposed absorber is based on the perfectly symmetrical pattern design of graphene, our absorbent structure is insensitive to the incident angle.

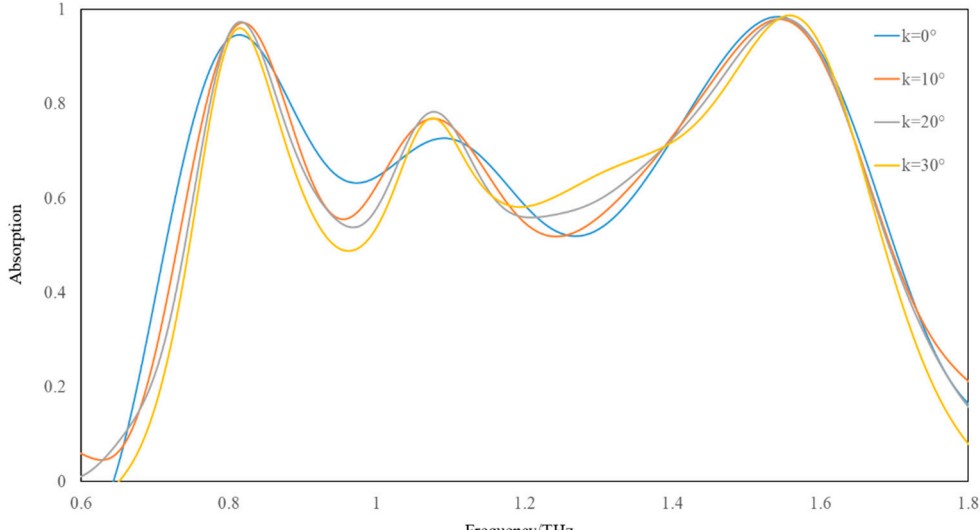

**Figure 7.** Absorption of the incident angle *k* of the incident electromagnetic wave propagating to the absorbing structure. The blue line indicates the absorption rate at *k* = 0°, the orange line indicates the absorption rate at *k* = 10°, the grey line indicates *k* = 20° and the yellow line indicates *k* = 30°.

## 4. Conclusions

We propose a triple-band perfect absorber based on a tunable graphene metamaterial that operates in the THz band. The nano-rings of different sizes resonate with electromagnetic waves of different frequencies, so that the entire absorber becomes a continuous broadband absorber. We have also achieved independent regulation of the perfect absorption of a single frequency, while giving our absorber a wide range of freedom to tune the absorption frequency. In addition, the wideband absorber has great potential in many aspects due to the perfect symmetry of its graphene structure.

**Supplementary Materials:** The following are available online at http://www.mdpi.com/2076-3417/10/5/1750/s1, Figure S1: The relationship of absorption and frequency, with and without ion-gel.

**Author Contributions:** X.J. and J.Y. conceived and designed the experiments; X.J. and Z.Z. performed the experiments; K.W., G.L. and J.H. analyzed the data; X.J. wrote the paper; J.Y., Z.Z. and K.W. edited the paper. All authors have read and agree to the published version of the manuscript.

**Funding:** This research received no external funding.

**Acknowledgments:** This work is supported by the National Natural Science Foundation of China (60907003, 61805278), the China Postdoctoral Science Foundation (2018M633704), the Foundation of NUDT (JC13-02-13, ZK17-03-01), the Hunan Provincial Natural Science Foundation of China (13JJ3001), and the Program for New Century Excellent Talents in University (NCET-12-0142).

**Conflicts of Interest:** The authors declare no conflict of interest.

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
