# Peer review of "A Triple-Band Hybridization Coherent Perfect Absorber Based on Graphene Metamaterial"

_applsci, doi:10.3390/app10051750_

Round 1

Reviewer 1 Report

The paper presents the design of a perfect absorber in THz range. The device is based on a metamaterial with structure unit composed of triple concentric graphene rings. The idea of tunability of the absorber mainly comes from the assumption that the Fermi level of all the graphene rings (or just a single ring) can be varied with the external gate. The whole idea might be interesting and inspiring for design of perfect absorbers. The topic is of interest and the novel ideas in the field are welcome.

Unfortunately, the important problem is the poor quality of English language used by the Authors. This makes significant parts of the work unclear and hard to understand. First of all, the language quality should be thoroughly improved.

I provide below some detailed remarks concerning the content of the work. The concern mainly the necessity of extending the description of the models used to obtain the numerical results.

In the schematic drawing, the inner and outer radii of the rings should be given clearly (or the outer radii and the width). The usage of "geometric parameter" is unclear, and this is a crucial point in description of the designed device.

The theory used to simulate the electric field pattern in section 3.2 (the results presented in Fig. 5) should be described in details - in the present version of the manuscript the origin of the results is unclear.

It should be described how the independent tuning of the Fermi levels of the rings might be achieved in the suggested device.

The theory used to calculate the data in Figs. 2-4 should be presented clearly – it seems not enough to give just the formulas for the conductivity. If FDTD is used, some computational details should be described.

In lines 93-94 the previous work is mentioned as explanation that the applied theory provides a good description - some reference should be cited here. The discussion after formula (5) seems to apply to array of nanoribbons - it should be carefully explained how it can be used in the present system.   What are the model parameters like epsilon or tau which enter the formulas 4 and 5?   What is denoted by the abbreviation SPP?

To conclude, the manuscript should undergo a major revision before it would be considered for publication in Applied Sciences.

Author Response

Thank you for your precious comments, which help a lot on improving this work. Here are responses according to your comments, and all revisions are marked in red in the revised manuscript.

Reviewer 2 Report

The authors have investigated the coherent absorptions of the triple hybridization absorber numerically by FDTD method. From the resutls, the authors have found two interesting results: one is the alteratoin effect of the Fermi level of the single graphene nanoring, and the second is the the angle-insenstivity thanks to the perfect symmetric structure of the graphene nanorings. The work provides interesting, but somewhat predictable numerical results.
Therefore, I believe that the most important part in this article is the simulation absorption spectrum by surface-based plasmonic resonance. It is however still lacking of detailed explanation of the numerical model scheme to capture how the authors have conducted the numerical simulations. The authors should explain more detail procedure for the numerical model.
The authors did not suggest any experimental data, which of course is predictable by numerical simulation but can significantly enhance the quality of the manuscript. The authors should at least rationalize their simulation work if they can’t add some convincing experimental data that can back up the theory.

Most of all, the authors should use some English correction services since there are so many grammar errors.

Author Response

(The authors gave the same response as above.)

Reviewer 3 Report

  In this paper, the authors present a triple-band hybridization light absorber based on patterned graphene coupled to an optical cavity. The idea is not novel and has been shown for the infrared frequencies in several publications:

Z. Fang, et al., ACS Nano, 2013. Z. Fang, et al., Nano Lett, 2014. A. Safaei, et al., Phy. Rev. B, 2017. A. Safaei, et al., ACS Nano, 2018.

The authors should cite all of these papers and apart from the frequency range, compare their method with that of the mentioned papers. 

  In addition, the paper represented Figure. 1 as the schematic of the proposed MPA. In the experiment, Ion-gel and SIO2 have optical phonons in that frequency range. But the authors did not include those phonons in their simulation. The presence of optical phonon has a significant role in the absorption spectrum. For more information, they can read the above-mentioned papers.

 Also, the authors mentioned, "the triple hybridization perfect absorber is angle-insensitive because of the perfect symmetry structure of the graphene nanorings". They should support this claim by showing the absorption spectra at different angles and a fixed Fermi energy.

  I believe this manuscript is not ready to publish at this stage and needs major revision.

Author Response

(The authors gave the same response as above.)

Round 2

Reviewer 1 Report

The Authors have revised extensively the manuscript according to the recommendations contained in the Review Reports and discussed the points raised by them. In response to the Reports, the quality of presentation was much improved. I would recommend the manuscript for publication in Applied Sciences in its present form.

Reviewer 2 Report

The authors have addressed all the questions, therefore, I recommend this manuscript for publication.

Reviewer 3 Report

I am convinced with the modifications, and I believe the manuscript is ready for publication.